# Mannose-Decorated Solid-Lipid Nanoparticles for Alveolar Macrophage Targeted Delivery of Rifampicin

**DOI:** 10.3390/pharmaceutics16030429

**Published:** 2024-03-20

**Authors:** Hriday Bera, Caizhu Zhao, Xidong Tian, Dongmei Cun, Mingshi Yang

**Affiliations:** 1Wuya College of Innovation, Shenyang Pharmaceutical University, Wenhua Road No. 103, Shenyang 110016, China; hriday.bera1@gmail.com (H.B.); zhaocaizhu961@163.com (C.Z.); tdealer@163.com (X.T.); 2Dr. B. C. Roy College of Pharmacy and Allied Health Sciences, Durgapur 713212, West Bengal, India; 3Department of Pharmacy, Faculty of Health and Medical Sciences, University of Copenhagen, Universitetsparken 2, DK-2100 Copenhagen, Denmark

**Keywords:** anti-tubercular drug, nanoformulations, targeted drug delivery, mannose

## Abstract

Alveolar macrophages play a vital role in a variety of lung diseases, including tuberculosis. Thus, alveolar macrophage targeted anti-tubercular drug delivery through nanocarriers could improve its therapeutic response against tuberculosis. The current study aimed at exploring the efficacy of glyceryl monostearate (GMS)-based solid-lipid nanoparticles (SLNs) and their mannose functionalized forms on the alveolar macrophage targeting ability of an anti-tubercular model drug, rifampicin (Rif). Rif-loaded SLNs were accomplished by the solvent diffusion method. These carriers with unimodal particle size distribution (~170 nm) were further surface-modified with mannose via Schiff-base reaction, leading to slight enhancement of particle diameter and a decline of drug loading capacity. The encapsulated Rif, which was molecularly dispersed within the matrices as indicated by their XRD patterns, was eluted in a sustained manner with an initial burst release effect. The uptake efficiency of mannose-modified SLNs was remarkably higher than that of corresponding native forms on murine macrophage Raw 264.7 cells and human lung adenocarcinoma A549 cells. Eventually, the mannose-modified SLNs showed a greater cytotoxicity on Raw 264.7 and A549 cells relative to their unmodified forms. Overall, our study demonstrated that mannose modification of SLNs had an influence on their uptake by alveolar macrophages, which could provide guidance for the future development of alveolar macrophage targeted nanoformulations.

## 1. Introduction

Alveolar macrophages are immensely important innate immune cells residing in the air–blood barriers of lungs [1]. They exhibit a strong phagocytic activity against invading pathogens. Similar to other macrophages, alveolar macrophages are non-immune, display a strong tissue specificity, maintain homeostasis and eliminate excess surfactants [2]. Thus, these are closely associated with the onset, progression and healing of many lung diseases, such as pulmonary fibrosis [3], lung cancer [4], asthma [5], tuberculosis [6] and so on. Tuberculosis is an infectious disease caused by *Mycobacterium tuberculosis* (*Mtb*) [7]. *Mtb* commonly enters into the lungs during respiration, and eventually it is taken up by alveolar macrophages, yielding phagosomes [8]. These phagosomes can then fuse with lysosomes to form phagolysosomes and kill the *Mtb* [9]. When *Mtb* cannot be effectively eliminated, it might interfere with the maturation of phagosomes, block the fusion of phagosomes and lysosomes by regulating the host signal transduction pathway and survive in the lungs for a long time [8,10,11].

Over several years, oral anti-tubercular drugs have been commonly used as a routine treatment strategy for tuberculosis [12]. Following oral administration, these drugs are widely distributed throughout the body, resulting in a very low drug concentration in the lungs, especially in the alveolar macrophages [13]. In order to achieve an effective inhibitory concentration, patients need a long-term oral administration of larger doses of anti-tubercular drugs, causing severe side effects such as liver toxicity, hypersensitivity and peripheral neurotoxicity [14]. A lower drug concentration in the alveolar macrophages could lead to the emergence of drug-resistant strains [15]. Serious side effects of anti-tubercular drugs and the existence of multidrug-resistant *Mtb* are the biggest challenges in the treatment of tuberculosis [12,13]. Nowadays, the targeted delivery of anti-tubercular drugs to the alveolar macrophages through pulmonary inhalation is considered an effective strategy for tuberculosis [16]. For instance, Shrivastava and co-workers formulated liposomes for macrophage targeted delivery dual anti-tubercular drugs [17]. Moreover, Pawde et al. developed chitosan-based scaffolds encapsulating clofazimine, which exhibited superior anti-mycobacterial activity compared to the pristine drug [18]. These could increase the local drug concentration in alveolar macrophages and effectively kill the pathogens retained there [13].

Recently, nanotechnology has been widely exploited as a promising strategy for alveolar macrophage targeted drug delivery [16]. Among various nano-size drug delivery systems, solid lipid nanoparticles (SLNs) have received overwhelming attention due to their distinct benefits like safety, biodegradability, affordable cost, etc. It is noteworthy that the mean particle size, charge, surface morphology, hardness, etc., might influence the cellular internalization of these particles [2]. In general, particles with suitable particle size range (100–200 nm), high positive or negative surface charges and suitable shape (such as spherical) with adequate hardness or stiffness would increase their uptake by the alveolar macrophages [1]. All these physiochemical properties of the SLNs are associated with their inherent compositions. Beside these factors, the cellular uptake of these nanosystems by alveolar macrophages could efficiently be improved by their surface modification with different ligands, leading to the enhancement of drug accumulation in the lung tissues [19]. Mannose is the most common alveolar macrophage-targeting ligand used to functionalize the surfaces of various nanosystems [20]. In our current work, glyceryl monostearate (GMS), a biocompatible and pharmaceutically acceptable emulsifier, would be utilized for the production of SLNs [21]. Stearyl amine (SA) could further be introduced to provide free amino groups onto the surfaces of SLNs [22]. The aldehyde groups of the ring-opened mannose could react with the amino groups of SA, creating mannosylated SLNs [23]. To our best knowledge, mannosylated SLNs composed of GMS and SA have not been formulated, nor even have their drug delivery properties been explored yet. The objective of our present study was to formulate rifampicin (Rif), a model anti-tubercular drug, loaded with GMS-based SLNs (Rif-SLNs) by the solvent diffusion method and investigate the impact of their surface modification with mannose on the alveolar macrophage targeting abilities through various in vitro investigations.

## 2. Materials and Methods

### 2.1. Materials

Glyceryl monostearate (GMS, Tianjin Bodi Chemical Co., Ltd., Tianjin, China), stearyl amine (SA, Shanghai Yien Chemical Technology Co., Ltd., Shanghai, China), rifampicin (Rif, Guangzhou Aichun Pharmaceutical Technology Co., Ltd., Guangzhou, China) D-(+)-mannose (Beijing SolarbioTechnology Co., Ltd., Beijing, China) and coumarin 6 (Beijing SolarbioTechnology Co., Ltd., Beijing, China), 4′,6-diamidino-2-phenylindole (DAPI, Dalian Meilun Bio-technology Co., Ltd., Dalian, China), 3-[4,5-dimethylthiazol-2yl]-2,5-diphenyl-tetrazolium bromide (MTT, Dalian Meilun Biotechnology Co., Ltd., Dalian, China) and Dulbecco’s modified eagle medium (DMEM, high glucose, Dalian Meilun Bio-technology Co., Ltd., Dalian, China) were utilized. Various other reagents and solvents used in this study were of analytical grade.

### 2.2. Formulation of Rif-Loaded SLNs (Rif-SLNs)

GMS-based SLNs were prepared by the solvent diffusion method according to previously reported protocol [24]. Briefly, GMS (15 mg) and Rif (4.5 mg) were dissolved in absolute ethanol (3 mL) and the mixture was heated at 75 °C. Subsequently, it was rapidly dispersed in Milli-Q^®^ water (30 mL) at 75 °C under stirring (600 rpm) for five min. The newly prepared dispersion was then cooled down to 25 °C to solidify the melted lipid droplets. Thereafter, the formulations were dialyzed (MWCO 12,000–14,000 Da, Shanghai Yuanye Bio-technology Co., Ltd., Shanghai, China) against Milli-Q^®^ water for 3 h to remove the free Rif and lyophilized.

### 2.3. Mannose Functionalization

SA (5 mg) was initially incorporated in the Rif-SLNs by adding SA with GMS, and similar steps were followed as described in Section 2.2. After the solidification of the SLNs, mannose functionalization was performed based on the previous report [20]. Precisely, mannose (50 mM) was dissolved in acetate buffer solution (pH 4.0) and heated in a water bath at 60 °C for 0.5 h. The resulting mannose solution (molar ratio of SA to mannose, 1:15) was added dropwise into the aqueous dispersion of SLNs. The mixture was magnetically stirred at 600 rpm for 72 h at 25 °C and subsequently dialyzed (MWCO 12,000–14,000 Da) for 0.5 h to obtain purified mannose modified Rif-SLNs (Rif-Man-SLNs).

### 2.4. Particle Size, PDI and Zeta Potential

All the samples were appropriately diluted with Milli-Q^®^ water for the determination of their particle size, PDI and zeta potential utilizing a Malvern Zetasizer Nano ZS instrument (Malvern Instruments Ltd., Malvern, UK).

### 2.5. Drug Loading

An accurately weighted quantity of freeze-dried SLNs (5 mg) was dispersed in methanol (1.2 mL) and the mixture was vortexed and heated at 60 °C for 2 min to completely dissolve the nanoparticles. It was then cooled at −20 °C for 10 min to precipitate the lipid. The supernatant was collected by centrifugation and assayed for Rif contents using a HPLC analysis method [25]. The drug loading (DL, %) and drug encapsulation efficiency (DEE, %) of the preparations were calculated according to the following formula [26].
$$DL\;\%=\frac{Actual\;drug\;loading}{Total\;mass\;of\;SLNs}\times 100\%$$
$$DEE\;\%=\frac{Actual\;drug\;loading}{Theoretical\;drug\;loading}\times 100\%$$

### 2.6. Appearance and Morphology

The freshly prepared SLNs were mounted onto the carbon films (Beijing Zhongjingkeyi Technology, Beijing, China) at room temperature and allowed to stand for 15 min to adsorb onto the films. The rest of the liquid was removed and dried overnight at room temperature. These specimens were then viewed under a transmission electron microscope (TEM, Hitachi, HT7700, Hitachi, Ltd., Tokyo, Japan).

### 2.7. Fourier Transform Infrared Spectroscopy (FT-IR)

Various samples were scanned using IFS 55 FTIR spectrometer (Bruker Corporation, Karlsruhe, Germany) within a wide wave number range (4000–400 cm^−1^).

### 2.8. Powder X-ray Diffractometer (P-XRD)

Powder X-ray diffraction patterns of different samples were recorded on a P-XRD (Haoyuan Instrument Co., Ltd., Dandong, China) at room temperature. It was operated within scanning angles of 3° to 40° (2θ) at a voltage of 40 kV and a current of 30 mA.

### 2.9. In Vitro Drug Release

Different formulations containing an equivalent amount of Rif (120 μg), which were calculated based on their drug loading, were introduced inside a membrane dialysis bag (MWCO 12,000–14,000 Da), which was placed into 20 mL of pre-warmed PBS (pH 7.4) containing vitamin C (VC, 200 μg/mL) and agitated in a water bath shaker (100 rpm, Suzhou Peiying Experimental Equipment, Suzhou, China) at 37 °C [27]. Samples (1 mL) were withdrawn at predefined time points and the cumulative drug release (%) was calculated based on the data obtained after HPLC analyses [25].

### 2.10. Cellular Uptake

#### 2.10.1. Confocal Laser Scanning Microscopy (CLSM)

Macrophage cells (RAW264.7) and lung cancer cells (A549) were suspended in DMEM medium supplemented with 10% fetal bovine serum, seeded in 6-well plates at a density of 3 × 10^5^ cells/mL and incubated in 5% CO_2_ incubator at 37 °C for 12 h. The cells were then treated with coumarin 6 (C-6)-loaded SLNs and incubated for 1 h and 4 h at 37 °C. Subsequently, the media were removed and the cells were washed three times with PBS. Thereafter, the cells were fixed with 4% paraformaldehyde (1 mL), stained with DAPI solution (10 μg/mL) for 5 min at room temperature and washed three times with PBS. The prepared samples were observed under CLSM (CLSM, Nikon C2 Plus, Nikon, Tokyo, Japan) [28].

#### 2.10.2. Flow Cytometer

The cellular uptake of SLNs was quantified by flow cytometer (BD FACSCalibur, BD Biosciences, Franklin Lakes, NJ, USA) [29]. The competitive effect of cellular uptake of mannose-modified SLNs was also assessed by pre-incubating the cells with free mannose (50 mM) for 1 h before treating them with the corresponding C-6-loaded SLNs. The treated cells were then examined by flow cytometer [20].

### 2.11. Cytotoxicity

The cytotoxic potential of the Rif-loaded SLNs against RAW264.7 and A549 cells was assessed by employing the MTT assay protocol [26]. Precisely, the cells (2000 cells/well) were seeded into 96-well plates and incubated in a 5% CO_2_ chamber (HERAcell 150i, Thermo Fisher Scientific, Shanghai, China) at 37 °C for 24 h. The cells were then treated with different concentrations of pure Rif, mannose-free and mannose-decorated SLNs and incubated for another 24 h. Subsequently, the MTT reagent was added to the wells. Following 4 h of incubation, the formazan crystals were dissolved in dimethyl sulfoxide (DMSO) and the absorbance of the plates was recorded on a multimode microplate reader (FLUOstar, BMG Labtech, Ortenberg, Germany) at 570 nm.

### 2.12. Statistical Analysis

Statistical analyses were performed using SPSS software (IBM SPSS Statistics 25.0) and differences were compared using an independent sample *t* test. Differences were considered as significant at a level of *p* < 0.05, highly significant at level of *p* < 0.01 and extremely significant at a level of *p* < 0.001.

## 3. Results and Discussion

### 3.1. Preparation and Physico-Chemical Characterization of SLNs

Tuberculosis is an extremely prevalent respiratory disease and the causative agents of tuberculosis often reside in the alveolar macrophages of lungs for a long period of time [7]. Recently, various carbohydrate-functionalized nanotherapeutics were exploited for macrophage targeted anti-tubercular drug delivery [30]. These could internalize the macrophages via overexpressed carbohydrate binding receptors and effectively suppress the invading microorganisms. The present study discussed the fabrication of mannose appended SLNs containing Rif and evaluated their effectiveness to accumulate the drug molecules inside the alveolar macrophages following endocytic uptake via the mannose receptor, a carbohydrate-binding receptor of the macrophage, for effective management of tuberculosis (Figure 1A) [22]. In other words, this could be accomplished through an active targeting process. On the contrary, passive targeting utilizes no homing device and uses only the natural distribution phenomenon of a carrier for drug targeting. In this context, Rif-loaded SLNs (Rif-SLNs) as control scaffolds were prepared using a solvent diffusion protocol, which involved the intense diffusion of solvent from the solvent–lipid phase into the surrounding aqueous phase followed by the evaporation of the organic solvent and solidification of lipid particles (Figure 1B) [26]. To afford Rif-Man-SLNs, SA was initially incorporated to provide amine terminated functionalities on Rif-SLNs. The coupling of mannose with these SLNs was then performed via Schiff bases chemistry, in which the aldehyde moieties of the ring-opened mannose could conjugate with the amino groups of SA present on the surfaces of drug-loaded SLNs (Figure 1C) [23].

The physical appearances of the drug-loaded SLNs were examined as compared to the corresponding blank templates (Figure 2A,B). Blank SLNs exhibited a light bluish opalescence due to the typical Tyndall effect [31], conferring the presence of abundant nanoparticles in the dispersions. On the other hand, drug-loaded SLNs appeared in different colors because of the presence of red-colored Rif molecules in the nanoparticles. Mannose-anchored and unmodified SLNs were further characterized in terms of their size, uniformity, zeta potential and morphology. The average hydrodynamic diameter of mannose-free SLNs was 167.7 ± 12.2 nm with a unimodal size distribution (PDI values, 0.19) (Table 1), while the mannose-modified SLNs exhibited a bigger particle size of 196.5 ± 15.5 nm [22]. Mannose conjugation could also enhance the PDI values (~0.320), indicating a higher heterogeneous characteristic of Rif-Man-SLNs compared to parent SLNs (Table 1) [32]. The zeta potential value of unmodified SLNs was about −24 mV. This was accredited to the existence of anionic charges on the outer layers of the SLNs owing to the presence of terminal carboxylic groups of the lipids [33]. The high zeta potential value of Rif-SLNs ensured the stability of the colloidal dispersions because of the electrostatic repulsions [34]. The functionalization of Rif-SLNs with mannose caused a highly positive zeta potential value (+42 mV) due to the existence of SA, a strong positively charged substance, in their nanoarchitectures [35]. This was consistent with the earlier reports [36]. Rif-SLNs exhibited an extremely high drug loading of 20.4 ± 0.7% with a DEE of 88.3 ± 3.2%, which was ascribed to the speedy configuration of rigid lipid nanoparticles, eventually restricting Rif leakage from the matrices [37]. The inclusion of SA into Rif-Man-SLNs conferred significantly reduced drug entrapment than the corresponding control matrices (Rif-SLNs) (*p*< 0.05). This was credited to the formation of leaky matrices (Rif-Man-SLNs) due to the presence of SA, causing extravasation of Rif molecules in the aqueous medium during their preparation. This was well collaborated with the previous report [20]. The TEM images of Rif-loaded SLNs showed spherical particles (Figure 2C,D). Their average size in TEM was slightly smaller compared to the DLS results, which could be due to their dehydrated state in the TEM grid.

### 3.2. FT-IR and XRD Analyses

FT-IR spectra of mannose, SA, GMS, their physical mixture, mannose-free and mannose-modified SLNs are compared in Figure 3A. In the spectrum of D-mannose, the broad signal at 3348.4 cm^−1^ and intense peak at 2916.6 cm^−1^ were attributed to its –OH stretching and –CH_2_ stretching vibrations, respectively. Moreover, the bands around 1630.5 and 1068.3 cm^−1^ were designated to the C=O stretching vibration of either alcohol or aldehyde groups in mannose [38]. On the other hand, the SA exhibited a lower intensity peak at 3361.9 cm^−1^, conferring an –NH_2_ stretching vibration. The signals at 2917.6 and 2849.4 cm^−1^ indicated –CH_2_ stretching of the long alkyl chains of SA molecules [39]. The GMS demonstrated various peaks at 3316.0 cm^−1^ (–OH stretching), 2915.7 cm^−1^ (–CH stretching), 1732.2 cm^−1^ (C=O stretching), 1470.7 cm^−1^ (–CH_3_ bending) and 1470.7 cm^−1^ (–CO bending) [40]. The FTIR spectrum of physical mixtures displayed simple superimposition of the typical bands of mannose, SA and GMS. Compared to mannose-free SLNs, the mannose-modified SLNs evidenced the existence of a –C=N stretching vibration peak at 1572.9 cm^−1^ and a bending vibration signal of –NH at 1413.1 cm^−1^, indicating mannose functionalization on the surfaces of Man-SLNs [20,22]. This was accomplished by the ring-opening reaction of the mannose and then the conjugation of its –CHO groups with the free –NH_2_ groups of SA situated on the surfaces of SLNs, resulting in the formation of Schiff’s bases (Figure 1C) [36].

The diffraction patterns of pristine components (Rif, GMS, SA and mannose), their physical mixtures and SLNs are portrayed in Figure 3B,C. The XRD curve of pure Rif revealed highly intense signals at 2θ of 9.9°, 11.9°, 13.8°, 14.5°, 16.4°, 18.6°, 19.5°, 21.4° and 26.1°, indicating its crystalline character [41]. The XRD patterns of GMS, SA and mannose also evidenced several intense and sharp diffraction peaks, which were indicative of their crystalline nature. The diffraction patterns of the physical mixtures were simply summations of the Rif and other components with identical sharp peaks and similar d (interplanar distance) values. The observed well-defined and sharp XRD peaks of pure Rif became less intense or disappeared in the case of Rif-SLNs and Rif-Man-SLNs. It confirmed a remarkable transformation of the pristine drug from its crystalline state to an amorphous form in the SLNs. In addition, Rif might strongly interact with component materials at the molecular level, affording a solid solution in the SLNs [42].

### 3.3. Drug Release

Rif could be hydrolyzed to its less soluble form, like 1-amino-4-methyl piperazine, under acidic conditions [43]. At pH values between 7.4 and 8.2, the drug molecule could be oxidized to an insoluble quinone derivative or a deacetylated form. VC, a small molecular weight antioxidant, plays a vital role in protecting against the oxidative damage of Rif by a variety of mechanisms [27]. Thus, VC was used as a stabilizer during the drug release study [44,45]. We initially investigated the stability of Rif in the release media (PBS, pH 7.4) containing variable amounts of VC. The stability of Rif in the PBS medium containing 200 μg/mL of VC was found to be better than that of the other two groups (Figure 4A). Based on this observation, PBS containing 200 μg/mL of VC was selected as the drug release medium in the subsequent experiments.

The drug release profiles of Rif-SLNs and Rif-Man-SLNs, as depicted in Figure 4B, indicate that the mannose modification had no significant impact on the drug release behavior. In both types of SLNs, the drug was rapidly released within the initial 6–8 h (~80%), and then slowly eluted until 48 h.

### 3.4. Cellular Uptake

The cellular uptake efficiencies of C6-loaded SLNs and Man-SLNs on Raw 264.7 and A549 cells at variable time points were investigated using CLSM and compared. The C6 was selected as a fluorescent tracker based on its structural resemblance with Rif. DAPI, a blue fluorescence dye, was employed to localize the cell nucleus [29]. Following treatment with these SLNs, the green fluorescence signals of C6 were located in the cytoplasm of the Raw 264.7 cells (Figure 5A), suggesting internalization of SLNs by the cells [28]. When the positively charged substance, SA, was introduced during the modification process, the zeta potential of the SLNs was drastically changed. However, the cellular uptake efficiency of C6-SA-SLNs remained closer to that of C6-SLNs, conferring insignificant effects of surface charges of SLNs on their cellular uptake behavior. On the contrary, the C6 labeled Man-SLNs portrayed an obviously stronger intracellular fluorescence signal on Raw264.7 cells than that of SLNs at either 1 or 4 h. The greater extent of uptake of the Man-SLNs by the Raw264.7 cells could be attributed to their mannose-mediated active targeting effects. The fluorescence intensity was augmented with increasing incubation time from 1 to 4 h, signifying that their cellular uptake was time-dependent [46]. The uptake studies of unmodified SLNs, SA-SLNs and Man-SLNs on mannose receptor overexpressing A549 cells [47] exhibited similar behavior to that of Raw264.7 cells (Figure 5B). The flow cytometry was also utilized to quantify the differences of the cellular uptake efficiencies of SLNs, SA-SLNs and Man-SLNs on Raw 264.7 and A549 cells, and the outcomes were consistent with the results of CLSM (Figure 5C–H).

We further attempted to examine the cellular uptake abilities of Man-SLNs on Raw 264.7 cells preincubated with or without free mannose by flow cytometry analyses (Figure 6). When the cells were preincubated with free mannose, the cellular uptake of Man-SLNs was remarkably reduced (*p* < 0.05) (Figure 6A,B). Free mannose could competitively inhibit the mannose receptors present on the cell surfaces and block the uptake of Man-SLNs by reducing receptor availability [20]. This implied that the cellular internalization of Man-SLNs could take place via mannose receptor-mediated endocytosis as well as non-specific adsorption-mediated trafficking [23].

### 3.5. Cytotoxicity

The cytotoxic potentials of pure Rif, Rif-SLNs and Rif-Man-SLNs were evaluated on two different cells (Raw264.7 and A549) by MTT assay protocol after 24 h of incubation (Figure 7) [26]. Pristine Rif and Rif-SLNs did not show cytotoxicity against Raw264.7 and A549 cells, with average cell viability around 100%. In contrast, Rif-Man-SLNs containing similar drug concentration ranges displayed varying degrees of cytotoxicity on both the cell lines. The cytotoxicity of Rif-Man-SLNs was increased with raising the loaded drug concentrations. Possibly, the presence of the positively charged substance SA in Rif-Man-SLNs could enhance their cytotoxicity [48]. In addition, the mannose-modified SLNs showed a greater cellular uptake efficiency, causing a higher cytotoxicity on Raw264.7 and A549 cells [49].

## 4. Conclusions

In this study, lipid GMS was successfully exploited to produce SLNs (~170 nm), which were further surface-modified with mannose. The particle size of the mannose-modified SLNs was increased slightly, their zeta potential became higher and the drug loading capacity was decreased relative to their native SLNs. The cellular uptake of Man-SLNs on Raw264.7 and A549 cells was significantly enhanced compared to unmodified scaffolds. The uptake of Man-SLNs was drastically inhibited when the cells were preincubated with mannose, indicating that their uptake was mediated through the mannose receptors. The mannose-modified SLNs showed a higher cytotoxicity at the effective antibacterial concentration of Rif due to their greater cellular uptake. Overall, the preliminarily experimental data evidenced the efficacy of mannose-modified SLNs to achieve greater alveolar macrophage targeting effects to combat tuberculosis.

## Figures and Tables

**Figure 1 pharmaceutics-16-00429-f001:**
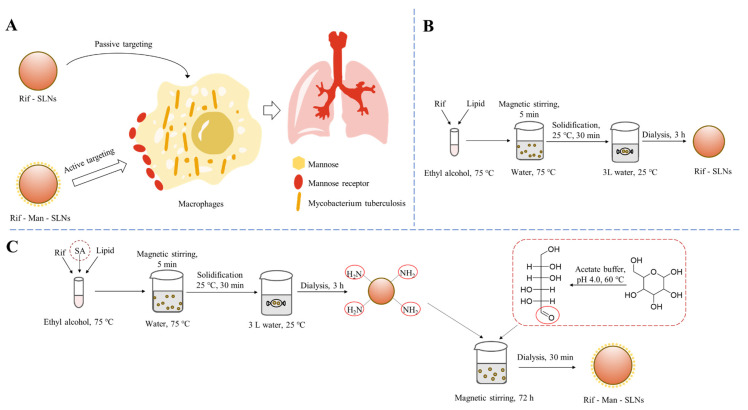
Schematic illustration of passive and active targeting of SLNs to macrophages (**A**). Preparation schemes of the Rif-SLNs (**B**) and Rif-Man-SLNs (**C**).

**Figure 2 pharmaceutics-16-00429-f002:**
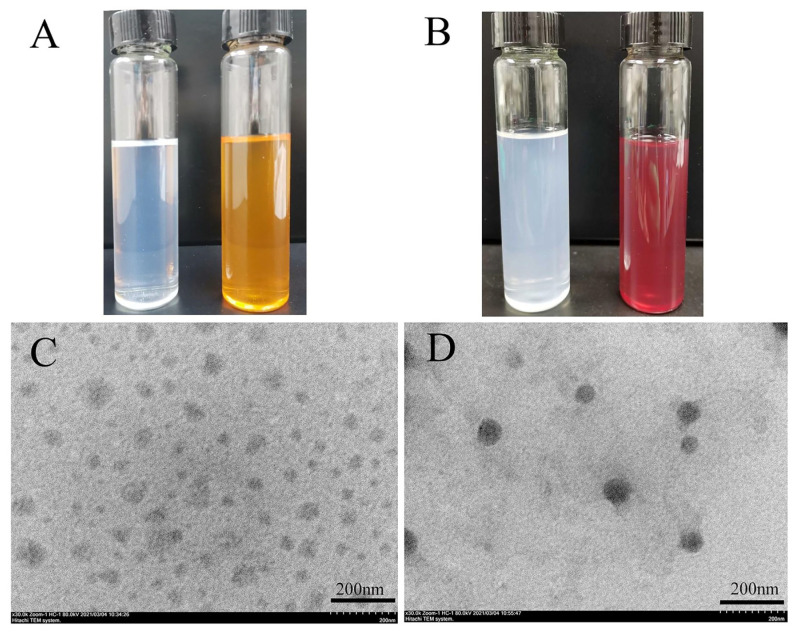
Physical appearances of blank SLNs (**left**) and Rif-SLNs (**right**) (**A**) and blank Man-SLNs (**left**) and Rif-Man-SLNs (**right**) (**B**). TEM images of Rif-SLNs (**C**) and Rif-Man-SLNs (**D**).

**Figure 3 pharmaceutics-16-00429-f003:**
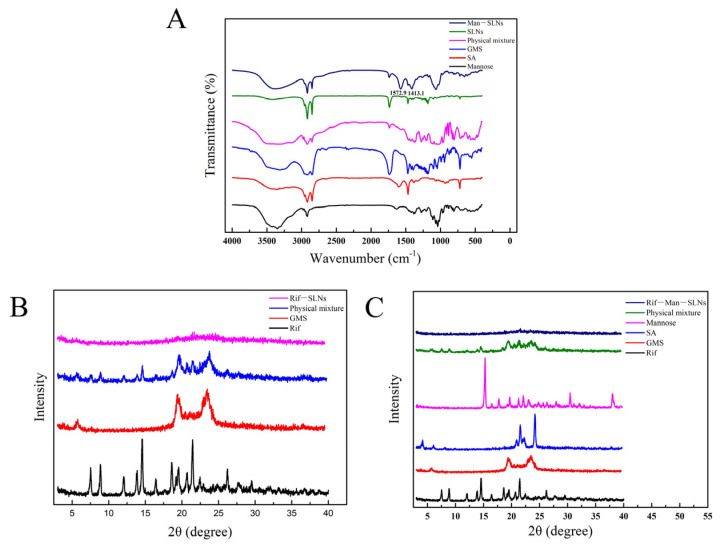
FT-IR overlay spectra of mannose, SA, GMS, physical mixture, SLNs, Man–SLNs (**A**). XRD patterns of Rif, GMS, physical mixture, Rif–SLNs (**B**) and XRD patterns of Rif, GMS, SA, mannose, physical mixture, Rif–Man–SLNs (**C**).

**Figure 4 pharmaceutics-16-00429-f004:**
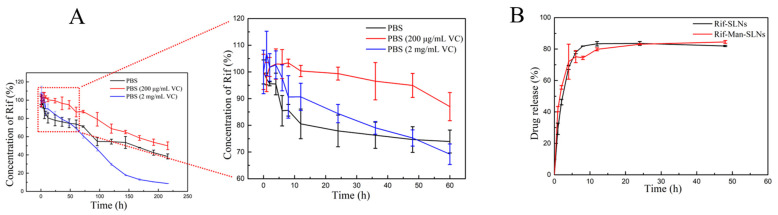
The stability of pure Rif in PBS media containing different amounts of VC (**A**). The in vitro release profiles of Rif from Rif-SLNs and Rif-Man-SLNs in PBS containing 200 μg/mL of VC (**B**) (*n* = 3, Mean ± SD).

**Figure 5 pharmaceutics-16-00429-f005:**
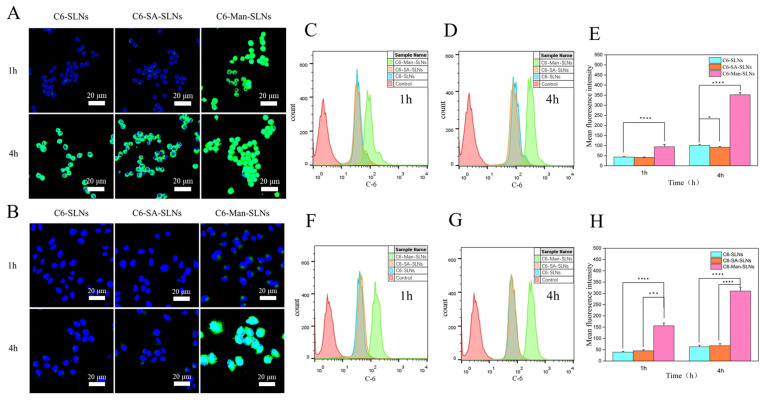
Cellular internalization of C6-SLNs, C6-SA-SLNs and C6-Man-SLNs on Raw 264.7 cells (**A**) and A549 cells (**B**) analyzed using CLSM. Cellular uptake efficiency of SLNs recorded by flow cytometry (**C**,**D**,**F**,**G**) and quantitative evaluation of their signals (**E**,**H**). Statistical significance levels are indicated as: * (*p* < 0.05); *** (*p* < 0.001); **** (*p* < 0.0001).

**Figure 6 pharmaceutics-16-00429-f006:**
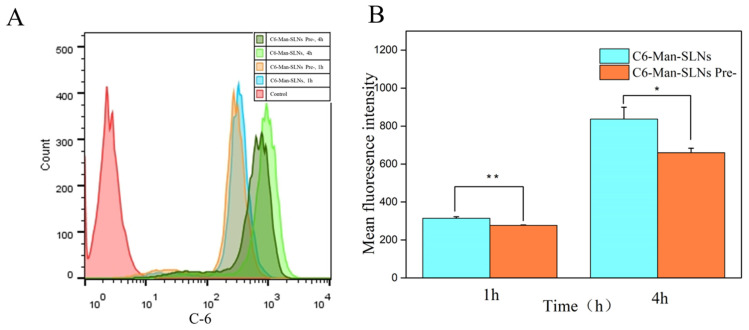
Flow cytometry studies of cellular uptake behavior of C6-Man-SLNs on Raw 264.7 cells with or without mannose preincubation (**A**) and their quantitative analyses (**B**). Statistical significance levels are indicated as: * (*p* < 0.05); ** (*p* < 0.01).

**Figure 7 pharmaceutics-16-00429-f007:**
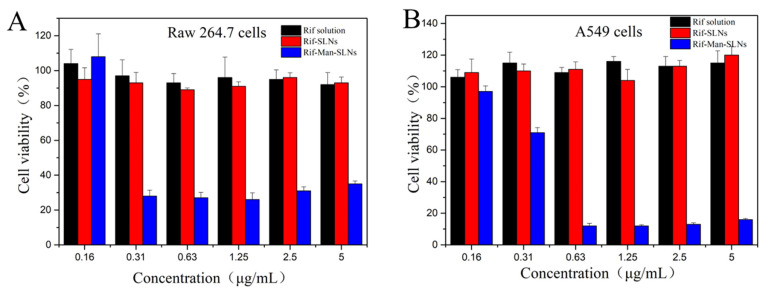
Cytotoxic potentials of Rif solution and different formulations on Raw 264.7 cells (**A**) and A549 cells (**B**) after 24 h of incubation (*n* = 5, Mean ± SD).

**Table 1 pharmaceutics-16-00429-t001:** Mean hydrodynamic particle size, PDI, zeta potential, DL and DEE of Rif-SLNs and Rif-Man-SLNs (*n* = 3, Mean ± SD).

Scaffolds	Particle Size (nm)	PDI	Zeta Potential (mV)	DL (%)	DEE (%)
Rif-SLNs	167.7 ± 12.2	0.191 ± 0.048	−24.4 ± 0.8	20.4 ± 0.7	88.3 ± 3.2
Rif-Man-SLNs	196.5 ± 15.5	0.320 ± 0.075	42.3 ± 1.1	1.0 ± 0.1	5.6 ± 0.4

## Data Availability

Data are contained within the article.

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
