# Peer review of "Mannose-Decorated Solid-Lipid Nanoparticles for Alveolar Macrophage Targeted Delivery of Rifampicin"

_pharmaceutics, 2024, doi:10.3390/pharmaceutics16030429_

Round 1

Reviewer 1 Report

Comments and Suggestions for Authors

The authors have written an article on mannose targeted slns for the delivery of rif. The article is interesting and overall well written; however there are some minor and major points that I think are critical for the publication of this article in pharmaceutics

-minor comments in the abstract the authors wrote >200 nm particles but they started around 170

-section 2.5 how much slns were weighed to be dissolved in 1.2mL MeOH

-section 2.9 120 ug was the calculated Rif amount or the theoretical

Major- if rif is so susceptabile to changes that will affect its biological activity, how do the others intend to use it when they are not coloading it into the SLNs? The point of the SLN is to protect the rif

-authors state c6 is used to act as a fluorescent tracker, but they do not perform the confocal with the rif and c-6 together?

i8s there an explanation why graphs 5c and h at 4 hours have such higher (but still very consistent) fluorescent readings than the controls in fig 6?

in the tox the authors state SA could lead to the toxicity. But what about the control of non loaded SLNS      MAN-SA-SLNs

The authors have written the article going directly from results to conclusion. I believe they should either add a discussion section or aument the discussion in the results to describe why these studies are different or more intersting than the previous works where they had already optimized the formation of these slns and linked them to mann-9. Therefore without mmore explanation and discusssion the only novelty here is the presence of Rif which often took a back seat to the man-9 effects

Comments on the Quality of English Language

the english is ok. Most errors are minor and just mispellings or minor article changes

Author Response

Reviewer 1:

The authors have written an article on mannose targeted slns for the delivery of rif. The article is interesting and overall well written; however there are some minor and major points that I think are critical for the publication of this article in pharmaceutics

-minor comments in the abstract the authors wrote >200 nm particles but they started around 170

Response: Authors express thanks to the reviewer. The particle size of Rif-SLNs has been revised to around 170 nm (Page no.: 2).

-section 2.5 how much slns were weighed to be dissolved in 1.2mL MeOH

Response: Authors are thankful to the reviewer for the comment. An accurately measured quantity of freeze-dried SLNs (5 mg) were weighed and dissolved in 1.2 mL methanol (Page no.: 6).

-section 2.9 120 ug was the calculated Rif amount or the theoretical

Response: Authors are grateful to reviewer for the insightful comment. Prior to the drug release study, the drug loading of different formulations was estimated. Various formulations containing equivalent amount of Rif of 120 μg were then used for the drug release study (Page no.: 7).

Major- if rif is so susceptabile to changes that will affect its biological activity, how do the others intend to use it when they are not coloading it into the SLNs? The point of the SLN is to protect the rif

Response: Authors are extremely gratified to the reviewer. In our current research work, rifampicin (Rif) was used as a model drug. The primary objective of our present study was to examine the effectiveness of SLNs in targeting the alveolar macrophages in the presence or absence of surface-decorated mannose (Page no.: 4).  

-authors state c6 is used to act as a fluorescent tracker, but they do not perform the confocal with the rif and c-6 together?

Response: Authors are enormously thankful for the reviewer. We agree that the co-localization of C6 and rifampicin could be considered. However, the C6 was selected as a fluorescent tracker based on its structural resemblance with Rif (Page no.: 14).  

i8s there an explanation why graphs 5c and h at 4 hours have such higher (but still very consistent) fluorescent readings than the controls in fig 6?

Response: Author thankfully acknowledge the reviewer comment. Fig. 5 illustrates the cellular internalization of C6-SLNs, C6-SA-SLNs and C6-Man-SLNs on Raw 264.7 cells and A549 cells, which were analyzed by CLSM and flow cytometry. Among various scaffolds, C6-Man-SLNs exhibited a greater extent of cellular uptake, which could be attributed to their mannose-mediated active targeting effects (Page no.: 14).   

On the other hand, Fig. 6 shows the results of flow cytometry studies of cellular uptake behaviour of C6-Man-SLNs on Raw 264.7 cells with or without mannose preincubation. When the cells were preincubated with free mannose, the cellular uptake of Man-SLNs was remarkably reduced. Free mannose could competitively inhibit the mannose receptors present on the cell surfaces and block the uptake of Man-SLNs by reducing receptor availability (Page no.: 15).   

in the tox the authors state SA could lead to the toxicity. But what about the control of non loaded SLNS      MAN-SA-SLNs

Response: Authors are highly thankful to the reviewer for the insightful comment.  The cytotoxic potentials of pure Rif, Rif-SLNs and Rif-Man-SLNs were evaluated on two different cells (viz., Raw264.7 and A549) and compared in Fig. 7. Among various groups, the cytotoxicity of Rif-Man-SLNs was increased with raising the loaded drug concentrations. Possibly, the presence of positively charged substance SA in Rif-Man-SLNs could enhance their cytotoxicity. In addition, the mannose-modified SLNs showed a greater cellular uptake efficiency, causing a higher cytotoxicity on Raw264.7 and A549 cells (Page no.: 16).

It is important to note that Rif-Man-SLNs contained stearyl amine (SA), which was introduced to provide free amino groups onto the surfaces of SLNs. The aldehyde groups of the ring-opened mannose could react with the amino groups of SA, affording mannosylated SLNs (i.e., Rif-Man-SLNs) (Page no.: 9). The SA free SLNs (i.e., Rif-SLNs) did not show cytotoxicity against Raw264.7 and A549 cells with average cell viability around 100% (Page no.: 16).

The authors have written the article going directly from results to conclusion. I believe they should either add a discussion section or aument the discussion in the results to describe why these studies are different or more interesting than the previous works where they had already optimized the formation of these slns and linked them to mann-9. Therefore without mmore explanation and discusssion the only novelty here is the presence of Rif which often took a back seat to the man-9 effects

Response: The valuable suggestions of the reviewer are highly appreciated.  The innovation point of the current research work has been revised in the introduction part and highlighted (Page no.: 4). Moreover, the discussion section of various results has been modified and highlighted (Page no.: 8-17).

Reviewer 2 Report

Comments and Suggestions for Authors

Thank you for submitting your paper to Pharmaceutics.
Overall, your paper is well-organize, well-written, and complete.
I have listed below a few minor improvements would make the paper more valuable to the reader.

____________________________________________

INTRODUCTION:
Well organized, clear, inclusive of other works in the field.
Line 71 begins to explain the contribution of this paper, however the language is a bit uncertain / unclear. The explanation of the contribution in lines 77-81 could be improved by adding brief information about the findings and clearly listing contributions of the work.

Materials and Methods: Excellent, very clearly written, very complete. Thank you!
____________________________________________

RESULTS:

Figure 1A: Can you more clearly describe passive targeting? It seems that the word passive only appears in the figure and in the figure caption, but not in the paper text.

Figure 2C and 2D: The TEM images do not strongly support the DLS measurements for diameter. Do the authors have any explanation for this

Figure 3: It is a little strange to see that the Rif-SLNs have none of the Rif peaks, which are very sharp on the Ric XRD data plot 3B. Likewise or Figure 3C. Is it just commonplace that formation of SLNs occludes or removes the XRD peaks for component materials from the XRD plots of the SLNs? I see line 257-259 clarify this data and include a good reference, but could the authors explain a bit further? Perhaps just 1 or 2 extra sentences, for the reader's benefit?

____________________________________________

Author Response

Reviewer 2:

Comments and Suggestions for Authors

Thank you for submitting your paper to Pharmaceutics.

Overall, your paper is well-organize, well-written, and complete.

I have listed below a few minor improvements would make the paper more valuable to the reader.

___________________________________________

INTRODUCTION:
Well organized, clear, inclusive of other works in the field.
Line 71 begins to explain the contribution of this paper, however the language is a bit uncertain / unclear. The explanation of the contribution in lines 77-81 could be improved by adding brief information about the findings and clearly listing contributions of the work.

Response: Authors express thanks to the reviewer for the valuable suggestions. The brief information about the findings and clearly listing contributions of the work have been included in the revised manuscript (Page no.: 4).

Materials and Methods: Excellent, very clearly written, very complete. Thank you!
____________________________________________

Response: Authors are grateful to the reviewer for the positive feedback.

RESULTS:

Figure 1A: Can you more clearly describe passive targeting? It seems that the word passive only appears in the figure and in the figure caption, but not in the paper text.

Response: The passive targeting has been described in the revised manuscript (Page no.: 9).

Figure 2C and 2D: The TEM images do not strongly support the DLS measurements for diameter. Do the authors have any explanation for this

Response: The reviewer’s comment is highly valued. The average size of SLNs in TEM images was slightly lesser as compared to the DLS results, which could be due to their dehydrated state in the TEM grid (Page no.: 10).

Figure 3: It is a little strange to see that the Rif-SLNs have none of the Rif peaks, which are very sharp on the Ric XRD data plot 3B. Likewise or Figure 3C. Is it just commonplace that formation of SLNs occludes or removes the XRD peaks for component materials from the XRD plots of the SLNs? I see line 257-259 clarify this data and include a good reference, but could the authors explain a bit further? Perhaps just 1 or 2 extra sentences, for the reader's benefit?

Response: We extremely appreciate reviewer’s suggestion. The observed well-defined and sharp XRD peaks of pure Rif became less intense or were disappeared in the case of Rif-SLNs and Rif-Man-SLNs. It confirmed a remarkable transformation of the pristine drug from its crystalline state to amorphous form in the SLNs.  In addition, Rif might strongly interact with component materials at molecular level, affording a solid solution in the SLNs (Page no.: 13).

Round 2

Reviewer 1 Report

Comments and Suggestions for Authors

The authors have satisfied this reviewer comments with their responses

Comments on the Quality of English Language

The english overall is fine with just minor issues that will be addressed in the proof processing